# Antimicrobial Activity of Snake β-Defensins and Derived Peptides

**DOI:** 10.3390/toxins14010001

**Published:** 2021-12-21

**Authors:** Nancy Oguiura, Poliana Garcia Corrêa, Isabella Lemos Rosmino, Ana Olívia de Souza, Kerly Fernanda Mesquita Pasqualoto

**Affiliations:** 1Ecology and Evolution Laboratory, Instituto Butantan, Sao Paulo 05503-900, SP, Brazil; poliana.correa@butantan.gov.br (P.G.C.); isabellarosmino@hotmail.com (I.L.R.); 2Development and Innovation Laboratory, Instituto Butantan, Sao Paulo 05503-900, SP, Brazil; ana.souza@butantan.gov.br; 3Alchemy–Innovation, Research & Development, University of Sao Paulo, Sao Paulo 05508-000, SP, Brazil; kfmpasqualoto@alchemydrugs.com.br

**Keywords:** β-defensins, snakes, antimicrobial activity, bioisosterism, peptides

## Abstract

β-defensins are antimicrobial peptides presenting in vertebrate animals. They participate in innate immunity, but little is known about them in reptiles, including snakes. Although several β-defensin genes were described in Brazilian snakes, their function is still unknown. The peptide sequence from these genes was deduced, and synthetic peptides (with approximately 40 amino acids and derived peptides) were tested against pathogenic bacteria and fungi using microbroth dilution assays. The linear peptides, derived from β-defensins, were designed applying the bioisosterism strategy. The linear β-defensins were more active against *Escherichia coli*, *Micrococcus luteus*, *Citrobacter freundii*, and *Staphylococcus aureus*. The derived peptides (7–14 mer) showed antibacterial activity against those bacteria and on *Klebsiella pneumoniae*. Nonetheless, they did not present activity against *Candida albicans*, *Cryptococcus neoformans*, *Trychophyton rubrum*, and *Aspergillus fumigatus* showing that the cysteine substitution to serine is deleterious to antifungal properties. Tryptophan residue showed to be necessary to improve antibacterial activity. Even though the studied snake β-defensins do not have high antimicrobial activity, they proved to be attractive as template molecules for the development of antibiotics.

## 1. Introduction

With the frightening advent of the global increase of microbial resistance to conventional antibiotics, the search for alternatives has become of utmost importance, and the industry, as well as the regulatory authorities, are realizing the potential of antimicrobial peptides. Since the last decade, antimicrobial peptides have been trialed in clinical phases [1].

In drug development, peptides’ properties have been considered more convenient due to their high affinity for the target and selective biological activities [2]. Bacterial resistance is a global health problem due to the indiscriminate use of antibiotics in humans, as well as in animals and in agricultural production that needs multipronged solutions [3]. To contribute to this campaign, many antimicrobial peptides have been discovered and some are in the clinical development phase [4]. For instance, molecules from innate immunity, such as cathelicidins present in snake venoms, have been reported as potentially active against some bacterial strains [5,6,7,8,9,10], including antibiofilm activity [11,12].

Snakes have developed many active peptides for predation and in their venom composition, many classes of proteins as phospholipases A_2_ (PLA_2_) [13], L-amino-acid oxidase (LAAO) [14], metalloproteases [15], cathelicidins [8] and crotamine [16,17,18] are also described as showing antimicrobial activity. Many attempts have been made to shorten these proteins and discover the active site [19,20,21,22]. Crotamine is a small basic myotoxin present in the venom of the rattlesnake *C. durissus terrificus* and has a β-defensin structure [23,24] and antimicrobial activity [16,17,18]. Crotamine and small basic myotoxins from this family are unique and only present in rattlesnake venom [23].

β-defensins from various vertebrates have been studied, and little is known about these molecules and the innate immunity in snakes. β-defensin-like genes with unknown functions have also been described in Brazilian snakes using a PCR approach [25,26,27]. In the current study, we evaluated the antimicrobial activity of snake β-defensins, both from rattlesnake venom (crotamine) and non-venom β-defensins. Its primary sequences were deduced from genomic sequences and synthesized, except for crotamine purified from rattlesnake venom. Moreover, the amino acid sequences were used to design shorter derived peptides, which can be obtained using simpler and more economical procedures. The snake β-defensins and derived peptides were assayed against microorganisms relevant to the snake’s biology and human health because they can cause opportunistic infections.

## 2. Results

The sequences of mature peptides were deduced from gene codifying sequences and synthesized, except crotamine purified from venom. They were tested in linear form because the linearization could not affect the antibacterial activity, and the linear form facilitates the development of shorter peptides. The alkylation was done to avoid the dimerization and the intramolecular cyclization of peptides with free thiol groups. Besides, the linear form facilitates the development of shorter peptides. Alkylated peptides were purified by high-performance liquid chromatography (HPLC) and analyzed by MALDI-TOF-MS.

The β-defensins were tested using a microbroth dilution assay. It consists of incubating the bacteria (4.105 colony-forming unit—CFU/mL) and the peptide (from 1 to 512 µg/mL) in liquid broth. After the incubation at 37 °C during the night, the bacterial growth was detected by spectrophotometry at 600 nm. If the measure is similar to the broth, it indicates bacterial growth. The molarity was calculated only to defensins that showed antibacterial activity to compare with the derived peptides. The β-defensins were tested against *B. jararaca* oral flora because it would be of biological and medical interest. In addition, we tested on *E. coli* ATCC 25922 and *M. luteus* that are common Gram-negative and Gram-positive bacteria used in antibacterial tests.

Table 1 shows the antibacterial effect of the snake β-defensins from venom (crotamine) and tissues indicating the minimal inhibition concentration that is the lowest concentration of peptide that results in no visible bacterial growth. It was possible to observe that the linearization of crotamine did not modify its antibacterial activity significantly. These β-defensins did not inhibit the bacterial growth of *Providencia rettgeri*, *Serratia marcescens*, *Morganella morganii*, and *Klebsiella pneumonia* up to 512 µg/mL, so the results were not shown in the table. *M. luteus* was the most sensitive bacterium to these peptides. The most active defensins present the highest cationic net charge, DefBm02 (+11), DefbBm03 (+10), DefbLm02 (+8), and crotamine (+7). On the contrary, the β-defensins DefbBju01 (+7), DefbBn02 (+2), and crotasin (−1) did not show antibacterial activity against any strains used in this study. In addition, the substitution in 32nd position, Arg (defbBm02) for Gln (defbBm03), has caused the loss of activity against *Staphylococcus aureus*.

The snake β-defensins crotamine, defbBm02, defbBm03, and defbLm03 were chosen to design the derived peptides for the next step due to their performance in the antibacterial assay. The crotamine 3D model was used as a template for designing the derived peptides because the 3D structure deposited in PDB presented a better resolution (1.70 Å). The software used to build the peptides has a tool to mutate, grow or delete amino acid residues, based on the sequence to be constructed. Furthermore, many algorithms clean the geometry to get a minimum energy structure to start the properties’ calculations. These peptides are presented in Table 2. They derived from the C-terminal portion of chosen β-defensins (Table 1). The design considered the bioisosterism strategy [29], for instance, the substitution of SH group (Cys) to OH group (Ser). Linear peptide constructions of different sizes (7 to 14 aa) were considered to verify which minor sequence would retain the antimicrobial activity. The peptides were commercially purchased, and their purities were tested by HPLC and mass spectrometry.

The primary structure does not illustrate a structure of interaction. The molecular surfaces translate the molecular form since the calculations consider a water molecule running through the van der Waals radius of each atom of the molecule. The molecular form property, consequently, is dependent on the 3D structure and can be used to visualize those differences.

The coordinates of the polypeptide crotamine from the Brazilian rattlesnake *C. durissus terrificus* [24], retrieved from Protein Data Bank (PBD ID 4GV5; resolution at 1.70 Å) [30], were used as starting geometry to construct the three-dimensional (3D) molecular models of the defensins derived peptides. The crotamine C-terminal portion, containing the fragments Cys30-Gly42 (13 aa), was used to build up the peptides’ 3D molecular models (see Material and Methods section). For instance, the Cys residues (SH group) were mutated into Ser (OH group), generating the peptide PS1, and the PS6 peptide was obtained by extracting Ser37-Gly42 from PS1. PS6 (7 aa) corresponds to the minor sequence assayed in this study. The molecular model of each peptide and its respective calculated solvent-accessible molecular surface are shown in Figure 1.

The peptide PS1 (SRWRWKSSKKGSG) is a bioisoster of the crotamine terminal portion (substitution of SH(Cys) to OH(Ser) group), and PS6 (SRWRWKS) shares the same sequence of PS1 from the 1st to 7th positions. The calculated solvent accessible surface area and molecular volume values were 1295. 61 Å^2^ and 1551.53 Å^3^ for PS1; 838.26 Å^2^ and 1013.05 Å^3^ for PS6, respectively. As mentioned above, PS6 is the minor peptide of this set. The peptides PS2 (SQMGRMSSRRRFGK) and PS4 (SQMGQMSSRRRFGK) have 14 amino acid residues and differ from one another only in the fifth position. PS2 has an arginine (positively charged) in the fifth position, whereas PS4 has a glutamine (polar; non-charged residue). The solvent-accessible surface area and molecular volume values found for PS2 were 1385.25 Å^2^ and 1672.60 Å^3^; and for PS4 were 1356.84 Å^2^ and 1633.52 Å^3^, respectively. The peptides PS3 (SGPGRRSSRRRK) and PS5 (SGPGRRSSRRRWK) share the same amino acid sequence from the 1st to 11th positions. PS5 has a tryptophan residue before the last residue, lysine (K). The solvent-accessible surface area and molecular volume values found for PS3 were 1179.24 Å^2^ and 1387.56 Å^3^; and for PS5 were 1355.29 Å^2^ and 1562.72 Å^3^, respectively. Concerning the solvent-accessible molecular surface or surface area values, the peptides can be classified in the following crescent order: PS6 < PS1 < PS3 < PS5 < PS4 < PS2. This property is related to both molecular shape and solvation process, which are important in the ligand–target recognition process.

These β-defensin-derived peptides were tested against bacteria using the microbroth dilution assay from 2.75 to 700 µg/mL. The results showed that these peptides were not active against *M. morganii* and *P. rettgeri* but presented various antibacterial properties against Gram-positive and Gram-negative bacteria (Table 3).

The derived peptides were also tested against fungi species using the classical resazurin microtiter assay plate method. It is based on the color change of the rezazurin dye that in blue indicates the absence of growth in at least 90%, and pink refers to microorganism growth. MIC is determined as the minimal concentration of the sample that can prevent the color change from blue to pink and refers to the inhibition of 90% of microorganisms. The samples did not inhibit the fungal growth and were considered ineffective until the highest assayed concentrations (250 µM) (Table 4).

Observing the molecular shapes (Figure 1) expressed by the obtained solvent surface area values, peptides PS2 and PS4 (14 aa) are the most related and showed similar activities. The replacement of amino acid residues at the fifth position, R5Q, to a polar non-charged residue showed to be deleterious to maintain the antibacterial activity against *E. coli* and *S. aureus* (Table 3). These results indicate the importance of the basic residue to retain the activity. On the other hand, the PS2 and PS4 inhibitory activity values on *M. luteus* were practically the same (Table 3), suggesting the R5Q substitution is not crucial for that interaction profile.

The PS3 (12 aa) and PS5 (13 aa) peptides differ from one another by the tryptophan (W) insertion at the 12th position, and the PS5 sequence modification can be visualized through the molecular shape (Figure 1). The sequence change has provided a significant improvement in the PS5 activity profile in comparison to PS3. PS5 has presented interesting MIC values against *E. coli* (55.2 µM), *S. aureus* (27.7 µM), *M. luteus* (13.9 µM), and *K. pneumoniae* (27.7 µM). Regarding *C. freundi*, the MIC value was higher (110 µM). The map of electrostatic potential (MEP) was calculated onto the molecular surface of both peptides, and the changes related to the electronic density distribution are presented in Figure 2. The insertion of tryptophan has contributed to exposing the positively charged region of PS5 (intense blue, lower electronic density distribution; arrow in Figure 2).

The PS1 peptide (13 aa) has presented interesting antibacterial activity against *E. coli* (56.5 µM), *S. aureus* (56.5 µM), and *M. luteus* (28.4 µM). As for PS5, regarding *C. freundi*, the MIC value was higher (113 µM). In comparison to PS1, the PS6 peptide (7 aa), having the sequence core SRWRWKS, did not show any activity against *C. freundi*. The minor sequence has retained indeed some antibacterial activity.

Unfortunately, none of the derived peptides have shown antifungal activity against the tested fungi, suggesting that the presence of Cys residues in the sequence is important to that kind of activity profile.

## 3. Discussion

The β-defensins were tested against *B. jararaca* oral flora because it would be of biological and medical interest. It is known that microorganism infection is one of the snakebite complications due to bacteria or fungi found in the snake’s oral cavity [32,33]. In Brazil, over 80% of snake envenomations are caused by *Bothrops* snakes, and around 10% of the snakebites evolve to infections [34]. Among bacteria isolated from abscesses, the most reported were *M. morganii*, *E. coli*, *Providencia* sp., *Klebsiella* sp. [34]. Interestingly, no β-defensins tested in this study showed antibacterial activity against *M. morganii* nor *K. pneumoniae*. Although bacteria were also isolated from the venom of *C. durissus terrificus* [35], envenomations by this species (*C. durissus terrificus*) do not usually cause infection or macroscopic necrosis in the bite site [34]. The toxins present in *C. durissus terrificus* venom: crotamine, PLA_2_, LAAO [36], and cathelicidins [22] could help the asepsis snakebite wound.

The linearization did not abolish the antibiotic activity of crotamine and the majority of snake β-defensins studied herein. The MIC against Gram-positive and -negative bacteria is shown in Table 1. The β-defensins presenting lower net charge values (crotasin, −1, defbBn02, +2) did not show antibacterial activity against any bacteria tested or showed a weak antibacterial effect with high MIC (defbLm01, +2). On the other side, the β-defensins defbBm02 (+11) and defbLm02 (+8) showed the best antibacterial activity against the bacteria *E. coli*, *C. freundii*, *M. luteus*, and *S. aureus* with MIC in the range of 0.8 to 12.8 µM. Interestingly, DefbBju, with a net charge of +7, did not inhibit any bacteria tested in this study, indicating that the 3D structure could be essential to its antibacterial activity. The Gram-positive bacteria *M. luteus* was the most sensitive to snake β-defensins while *S. aureus* was inhibited only by defbBm02 and defbLm02. The Gram-negative bacteria *E. coli* and *C. freundii* were inhibited by β-defensins with a net charge higher than +7. The basicity of AMPs is related to antibacterial activity, the higher the positive charge higher the inhibition of bacteria growth, but there are indications that the disulfide bridges modulate the antimicrobial activity [37] and the balance of hydrophilic and hydrophobic surfaces [38]. The net charge of β-defensins was thoroughly discussed by Huang et al. [39]. The activity of linear crotamine corroborates the antibacterial activity of reduced crotamine [17] as well as antifungal activity [18].

Although *S. aureus* is sensitive to several toxins from snake venoms such as PLA_2_ [13,40,41,42,43,44,45], LAAO [46,47,48], and cathelicidins [8]; crotamine did not inhibit the growth of this bacterium; however, the β-defensins from snakes defbLm02 and defbBm02 showed antibacterial activity with MIC of 32 and 64 µg/mL respectively. Although native crotamine did not show activity against *S. aureus*, fragments of this toxin inhibited the growth of this bacterium [18].

Although *K. pneumoniae* is inhibited by cathelicidins [5,49,50], PLA_2_ [40,41,42,43,45], and LAAO [51,52,53], no β-defensin tested in this work showed antibacterial activity against this species. The same was true for *S. marcensens*, sensitive to LAAO [53,54], but not to cathelicidins [5,50]. *M. morganii*, sensitive to the venom of *Montivipera bornmuelleri* [55], was also resistant to snake β-defensins and crotamine.

As most of the bacteria tested were isolated from the oral flora of *B. jararaca*, the lack of antibacterial activity or even weak activity of these molecules may be because these β-defensins only control the coexistence of animals with these microorganisms and the tested snake β-defensin could have other functions in the animal. Many biological functions are described to antimicrobial peptides such as modulation of inflammatory responses, wound healing, angiogenesis promotion [56], regeneration of a lizard tail [57,58], and sperm function in mammals [59], but these biological activities depend on the 3D structure of the β-defensins [60].

The goal of the design of short peptides based on our β-defensins is to get more accessible and cheaper manufacturing as the advantages of small molecules [61]. Peptides P1 to P6 (14 to 7 residues) were tested against bacteria and fungi.

The use of β-defensin fragments caused the decrease of antibacterial activity (see MIC values in Table 3), which was expected since the isolated fragments did not have the same behavior as a conformational organized protein structure. However, it also led to positive results: PS1 showed antibacterial activity against *C. freundii* and *S. aureus* differently from crotamine, as well PS5 that inhibited the growth of *K. pneumoniae*, unlike defbLm02. The decreased antibacterial activity presented by smaller analogs in comparison to the original was also reported to the β-defensin HBD3 [60,62]. A probable cause of the decrease may be the proteolysis that these peptides may be subject to since they were not structurally protected [63]. The peptide with the highest performance was PS5, its only difference to PS3 being an introduction of Trp between basic residues, which may have improved the charge net facilitating the interaction with the bacterial membrane, as happened with the decamer derived from HBD28 [64]. Wessolowski et al. [65] observed that introducing Trp residue in the sequence increases the antibacterial activity, and the cyclization increases the antibacterial activity and the selectivity of peptides. Interestingly, the substitution of Arg for Gln (defbBm02 to defbBm03) did not alter the activity against *E. coli*, *M. luteus*, and *C. freundii*, but it seemed essential to inhibit *S. aureus* by defbBm02. On the other hand, the same substitution was critical for PS4 (derived from defbBm03) to inhibit *S. marcenses*.

It is known that fungi are sensitive to toxins from snake venoms, and LAAO [48], PLA_2_ [43], cathelicidins [5,50], and crotamine [18] were already evaluated on *C. albicans*. Other sensitive fungi are *Aspergillus niculans* [5] and *Cryptococcus neoformans* [18]. In contrast, *C. neoformans* was resistant to cathelicidins [51]. Despite the crotamine fragment (residues 27–39) having inhibited the growth of *C. neoformans* [18], the substitution of Cys by Ser in crotamine fragments was deleterious for the antifungal activity [66] in the same way this substitution also abolished the activity against *C. albicans* [67]. Unfortunately, this was true to all the peptides designed in this work showing that this substitution is deleterious to antifungal activity.

We observed that the C-terminal segment of β-defensins is essential to antimicrobial activity, as also observed by Mandal et al. [68]. Additional molecular modifications seem to be necessary to improve the conformational arrangement and to aid the establishment of the structure–property/activity relationships. Based on that, novel promising peptides can be designed to have better antibacterial activity (higher potency). However, to improve the antibacterial activity of derived peptides, it is necessary to protect them from proteolysis by chemical alteration, C-terminal amidation [64], or cyclizing the peptide by disulfide bonds [63], as also the inclusion of Trp in the sequence to ameliorate the balance between hydrophobicity and hydrophilic or to stabilize the peptide structure favoring the membrane interaction. The substitution of Cys residue by Ala should be also considered, since it has been reported to increase the antibacterial activity of short peptides, as well as the substitution of Leu and Ile by Trp [69].

A feature of the β-defensin family is the 3D structure conserved by the disulfide bridges of the cysteine motif and a significant variation of amino acid sequences [70]. These molecules are good templates for development studies since biological targets have already been selected by nature [71].Furthermore, its mechanism of action, which causes the rupture of the bacterial membrane and can bind to different targets as DNA, makes it difficult for the bacteria to develop resistance [56,70].

In summary, the studied snake β-defensins do not have optimum antimicrobial activity, but they proved to be attractive as template molecules for the development of antibiotics. Our data indicate that the short peptides show specific activity on prokaryote cells but not on eukaryote cells. Based on that, the snake β-defensins C-terminal portion, if optimized, could be indeed used as a bioactive agent.

## 4. Material and Methods

Twelve peptides (GenBank Accession number is shown in Table 5) were synthesized by Biomatik (Wilmington, NC, USA) and coded as crotasin, DefbBd03, DefbBj01, DefbBju01, DefbBm02, DefbBm03, DefbBn02, DefbLm01, DefLm02, DefbPm, DefbTs and hBd02. The amino acid sequences were deduced from genes and are presented in Table 3. The synthetic peptides were treated with 45 mM DTT by 15 min at 50 °C [72] followed by alkylation using 100 mM IAA (iodoacetamide) 15 min, at room temperature. Alkylated peptides were purified by high performance liquid chromatography (HPLC) and analyzed by MALDI-TOF-MS. This step was performed at the Laboratory of Applied Toxinology—Instituto Butantan with the supervision of Dr. Pedro I. da Silva Jr.

Crotamine was purified from the *C. durissus terrificus* venom (purchased from CEVAP, Botucatu, SP, Brazil) and purified as described [73]. Briefly, crotamine was purified from crude venom by size exclusion on a Sephacryl S200 column (GE Healthcare, Uppsala, Sweden), followed by cation exchange chromatography on a 1-mL Resource S on FPLC system (Akta Purifier System, GE Healthcare). The identity and purity of crotamine were confirmed by MS analysis. The reduction and alkylation were proceed as described above.

Crotamine, defbm02, defBm03, and defbLm02 were chosen to design the short peptides due to the best antibacterial activity among the β-defensins tested. Previous studies have indicated the C-terminal of the human β-defensin HBD3 and crotamine as the mandatory region of antimicrobial activity [18,22,60,66,67].

The short peptides, derived from β-defensins (Table 2), were designed applying the bioisosterism strategy [29]; for instance, the substitution of SH group (Cys) by OH group (Ser), regarding the C-terminal portion of the β-defensins (crotamine, defbBm02, defbLm02, defbBm03). Bioisoster groups or substituents share chemical or physical similarities, producing similar biological properties. It was considered linear peptide constructions having different sizes (7 to 14 aa) to verify which would be the minor sequence able to retain the defensins biological activity exploited. The designed peptides were synthesized by GenOne Biotechnologies. The amino acid sequences are presented in Table 1.

### 4.1. Molecular Modeling and Molecular Properties Calculation

The coordinates of the polypeptide crotamine from the Brazilian rattlesnake *C. durissus terrificus* [24] were retrieved from Protein Data Bank (PBD ID 4GV5; resolution at 1.70 Å) [30] and used as starting geometry to extract the fragment Cys30-Gly42 (13 aa), which were employed to build up the peptides’ 3D molecular models. The Cys residues were mutated into Ser, generating the peptide PS1. The PS6 peptide was obtained by extracting Ser37-Gly42 from PS1. PS6 (7 aa) corresponds to the minor sequence designed and assayed in this study. The other fragments were constructed using the tool build-mutate or build-grow available in the Discovery Studio Visualizer 4.0 software (Accelrys Software Inc., 2005–2013). The geometries were optimized, and partial atomic charges were assigned using the CHARMM force field [63], included in Discovery Studio (Accelrys Software Inc., 2005–2013).

Furthermore, the electrostatic potential (EP) property was calculated for the PS3 and PS5 peptides to visualize the changes in electronic density distribution concerning the amino acid substitution patterns (PS5 has one more residue, Trp, in comparison to PS3). The charges from electrostatic potential using a grid-based method (CHELPG [64]) were calculated employing the ab initio method Hartree-Fock/3-21G* basis set (Gaussian 03W software; Gaussian, Inc., Pittsburgh, PA, USA, 2003). The EP maps were calculated onto the peptides’ molecular surfaces using GaussView 05 software (Gaussian, Inc., Pittsburg, PA, USA, 2002–2008). The interpretation of EP maps is based on a color scheme, where regions having higher electronic density distribution are presented as an intense red color (negatively charged), whereas regions with lower electronic density distribution are shown as an intense blue color (positively charged). Since the EP property has been calculated onto the PS3 and PS5 molecular surfaces, their molecular shapes were also assessed. Moreover, the molecular volume (intrinsic molecular property) of each peptide considering the van der Waals radii was also calculated employing Discovery Studio Visualizer 4.0 software (Accelrys Software Inc., 2005–2013). Of note, the molecular shape and electronic properties are among the primary molecular properties in the ligand–receptor recognition process.

### 4.2. Antibacterial Activity

The antibacterial activity of the alkylated peptides was tested against Gram-negative (G−) bacteria (*Klebsiella pneumonia*, *Serratia marcescens*, *Morganella morganii*, *Providencia rettgeri*, *Citrobacter freundii*, *Escherichia coli* ATCC 25922 and Gram-positive (G+) bacteria (*Micrococcus luteus* A270 and *Staphylococcus aureus*) using microbroth assay [28]. In a 96-well plate, 90 µL of 10% tryptone soy broth (TSB) with 4 × 10^5^ colony-forming unit (CFU/mL) were mixed with serial twofold dilutions of β-defensins duplicate in concentrations from 512 to 1 µg/mL. The derived peptides coded as PS1, PS2, PS3, PS5, and PS6 (Table 1) were tested against these bacteria in serial twofold dilutions with concentrations bellow 700 µg/mL. After overnight incubation at 37 °C, the turbidity of the bacterial culture was read at 600 nm (Epoch microplate reader, Biotek). The minimal inhibitory concentration (MIC) of each peptide was determined as the lowest concentration that results in no visible bacterial growth. The MIC resulted from three independent experiments. The bacteria *K. pneumonia*, *S. marcescens*, *M. morganii*, *P. rettgeri*, *C. freundii*, *M. luteus* A270 and *S. aureus* were isolated from *Bothrops jararaca* buccal flora and kindly provided by Dr. Márcia R. Franzolin (Laboratory of Bacteriology—Instituto Butantan). *M. luteus* was kindly provided by Dr. Pedro I. da Silva Jr. (Laboratory of Applied Toxinology—Instituto Butantan).

### 4.3. Antifungal Activity

The antifungal activity was evaluated by the microdilution assay as previously described [31] on clinical strains of *Candida albicans* (IOC 4525), *Cryptococcus neoformans* (IOC 4528), *Trichophyton rubrum* (IOC 4527) and *Aspergillus fumigatus* (IOC 4526). The fungi were cultivated in potato dextrose agar at 28 °C according to the Clinical and Laboratory Standards Institute recommendations [75]. Fungal suspensions were prepared in RPMI 1640 culture media; the CFU/mL was adjusted to 0.5–2.5 × 10^3^ for yeasts (*C. albicans* and *C. neoformans*) and 0.4–5 × 10^4^ CFU/mL for filamentous fungi (*T. rubrum* e *A. fumigatus*), by comparison with a standard curve previously stablished in our laboratory.

The peptides PS1, PS2, PS, PS3, PS4, PS5 and PS6 at 1 mM in acetic acid at 0.01% were diluted in RPMI 1640 culture media at concentrations ranging from 1 to 250 μM. Amphotericin B (AMB) at concentrations below 15 μg/mL (16 μM) and acetic acid from 0.0002 to 0.0025% was used as control.

The fungi suspensions were added to a 96 well plate (100 µL/well) and the samples (100 µL) in different concentrations were added to each well. The plate was incubated at 28 °C for 24–72 h, and 24 h before the end of the assay 25 µL of the rezazurin dye at 0.02% were added to each well. The minimal inhibitory concentration (MIC90) was defined as the lowest concentration that prevents the rezazurin’s change in color from blue to pink due to the inhibition of at least 90% of the microorganism’s growth. The assays were performed in three independent experiments.

## Figures and Tables

**Figure 1 toxins-14-00001-f001:**
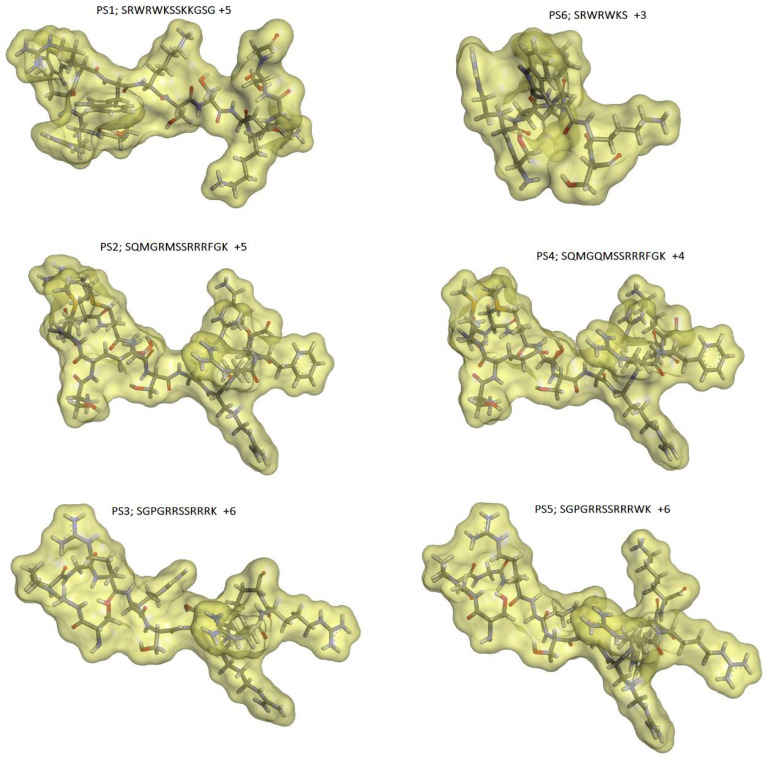
Solvent accessible molecular surfaces using a probe of 1.4 Å (water molecule radius). The peptides are presented in stick models, where hydrogen atoms are in white, oxygens in red, nitrogen atoms in blue, sulfur in orange, and carbon atoms in gray. The molecular surfaces are translucid and presented in yellow color (Discovery Studio Visualizer 4.0 program, Accelrys Software Inc., 2005–2013, San Diego, CA, USA).

**Figure 2 toxins-14-00001-f002:**
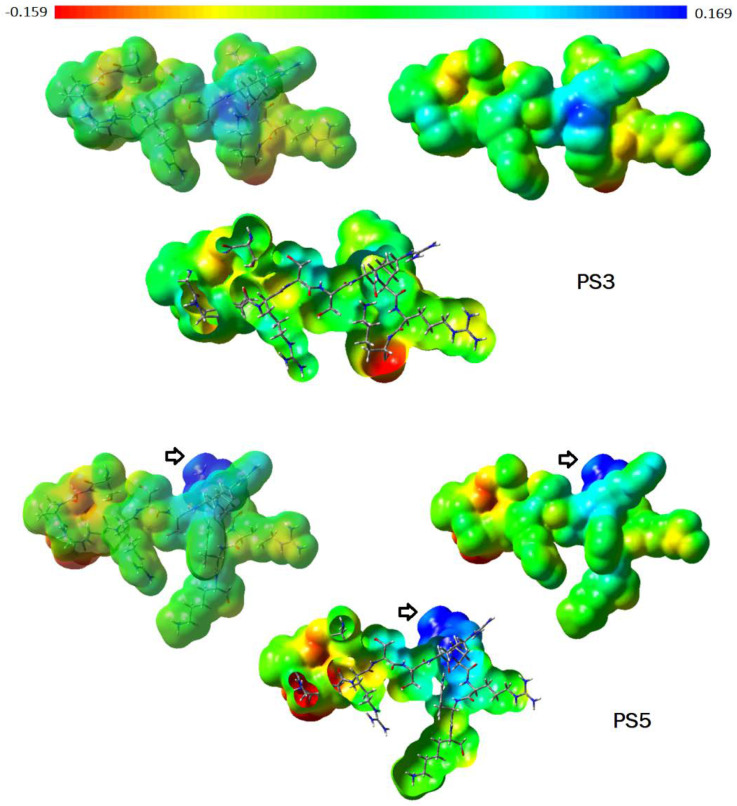
Maps of electrostatic potential (MPEs) were calculated onto the molecular surface of PS3 and PS5 peptides. Regarding the color range, higher electronic density distribution regions are displayed as intense red color (−0.159), and lower electronic density distribution regions are shown in intense blue color (+0.169) (Gaussian 03W, Gaussian, Inc., Pittsburgh, PA, USA, 2003; GaussView 05, Gaussian, Inc., Pittsburgh, PA, USA, 2002–2008).

**Table 1 toxins-14-00001-t001:** Antibacterial effect expressed by the minimal inhibitory concentration (MIC) of snake β-defensins on bacteria.

	MIC (µM)			
Samples	*E. coli*	*M. luteus*	*C. freundii*	*S. aureus*
Crotamine (+7)	13.3	1.7	6.7	>105
Linear crotamine	26.6	1.7	26.7	>105
DefbBm02 (+11)	3.2	0.8	1.6	12.8
DefbBm03 (+10)	1.6	1.6	3.2	>103
DefbBd03 (+6)	>115	0.9	>115	>115
DefbBj01 (+4)	>105	3.2	>105	>105
DefbBju01 (+7)	>111	>111	>111	>111
DefbBn02 (+2)	>113	>113	>113	>113
DefbLm01 (+2)	>113	28.4	>113	>113
DefbLm02 (+8)	3.4	0.9	3.4	6.8
DefbPm (+6)	>109	6.8	>109	>109
DefbTs (+3)	>122	15.3	>122	>122
hBd02 (+6)	15.6	2	>124	>124
Crotasin (−1)	>108	>108	>108	>108

The MIC was determined to be the lowest concentration, resulting in no visible bacterial growth in microbroth dilution assay [28]. The assays were performed in three independent experiments. All the β-defensins tested did not inhibit the growth of *Providencia rettgeri*, *Serratia marcescens*, *Morganella morganii*, and *Klebsiella pneumonia*.

**Table 2 toxins-14-00001-t002:** Peptides derived from β-defensins applying the bioisosterism strategy.

Peptide	Derived from	Sequence	Hidrofobicity (Kcal/mol)	Net Charge	MW (Da)
PS1 (13 aa)	Crotamine	SRWRWKSSKKGSG	+19.88	+5	1549
PS2 (14 aa)	defbBm02	SQMGRMSSRRRFGK	+19.34	+5	1683
PS3 (12 aa)	defbLm02	SGPGRRSSRRRK	+23.57	+6	1399
PS4 (14 aa)	defbBm03	SQMGQMSSRRRFGK	+18.30	+4	1655
PS5 (13 aa)	PS3	SGPGRRSSRRRWK	+21.48	+6	1585
PS6 (7 aa)	PS1	SRWRWKS	+11.06	+3	1005

**Table 3 toxins-14-00001-t003:** Antibacterial effect expressed by the minimal inhibitory concentration (MIC) of short peptides derived from snake β-defensins on bacteria.

	MIC (µM)				
Peptides	*E. coli*	*S. aureus*	*M. luteus*	*K. pneumoniae*	*C. freundii*
PS1	56.5	56.5	28.4	>452	113
PS2	415	415	26.1	>415	>415
PS3	>500	125	31.5	>500	>500
PS4	>423	>423	26.6	>423	>423
PS5	55.2	27.7	13.9	27.7	110
PS6	697	349	43.8	>697	>697

The MIC was determined as the lowest concentration that results in no visible bacterial growth in microbroth dilution assay [28]. The assays were performed in three independent experiments. All the short peptides tested did not inhibit the growth of *Providencia rettgeri, Serratia marcenses*, and *Morganella morganii*.

**Table 4 toxins-14-00001-t004:** Antifungal effect expressed by the minimal inhibitory concentration (MIC90) of short peptides derived from snake β-defensins on yeasts and filamentous fungi.

	MIC90 (µM)			
Peptides	*Candida albicans*	*Cryptococcus neoformans*	*Trichophyton rubrum*	*Aspergillus fumigatus*
PS1	>250	>250	>250	>250
PS2	>250	>250	>250	>250
PS3	>250	>250	>250	>250
PS4	>250	>250	>250	>250
PS5	>250	>250	>250	>250
PS6	>250	>250	>250	>250

MIC The minimal inhibitory concentration (MIC90) was defined as the lowest concentration that prevents the rezazurin’s change in color from blue to pink due to the inhibition of at least 90% of the microorganism’s growth [31]. The assays were performed in three independent experiments.

**Table 5 toxins-14-00001-t005:** β-defensins sequences deduced from genes and biochemical characteristics.

Peptide/ Snake	GenBank Accession Number	Amino Acid Sequence	Hidrofobicity Kcal/mol	Net Charge at pH 7	MW Da
Crotamine/ *C. durissus*		YKQCHKKGGHCFPKEKICLPPSSDFGKMDCRWRWKCCKKGSG	+49.14	+8	4886
Crotasin/ *C. durissus*	AF250212	QPQCRWLDGFCHSSPCPSGTTSIGQQDCLWYESCCIPRYEK	+28.91	−1	4708
defbBd03/ *B. diporus*	KC117160	QPECLRQGGMCRPRLCPYVSLGQLDCQNGHVCCRKKPRK	+37.08	+5	4456
defbBj/ *B. jararaca*	KC117163	QEECLQQGGFCRLIRCPFGYDSLEQQDCRKGQRCCIRKPRK	+45.05	+4	4874
defbBju/ *B. jararacussu*	KC117165	QRRCHQKGGMCLPGPCPPGYDSLGQQDCRRGQKCCIKRFGK	+43.85	+7	4591
defbBm02/ *B. mattogrossensis*	KC117167	QRRCRQRRGICRPRPCPPENFSLGRLDCQMGRMCCRRRFGK	+39.97	+11	4978
defbBm03/ *B. mattogrossensis*	KC117168	QRRCRQRRGICRPRPCPPENFSLGRLDCQMGQMCCRRRFGK	+38.93	+10	4950
defbBn02/ *B. neuwiedi*	KC117169	QPECCQEGGICHSKQCPLGYSSLGRLDCQLGQRCCIRIFGK	+33.65	+2	4513
defbLm01/ *L. muta*	KC117171	QEWCRGLGGFCSFYQCRPGHDLGPQDCWPERRCCRWGK	+33.69	+2	4515
defbLm02/ *L. muta*	KC117172	QGQCHQQRGRCFLHQCPLSHYFLGRLDCGPGRRCCRRRK	+36.90	+8	4655
defbPm/ *P. mertensis*	KX664436	QRICLGGRGFCHSTPCPRSTIDYGKKDCWGSLRCCEPKRPGK	+42.13	+6	4695
defbTs/ *T. strigatus*	KX664429	QDLCHNLGGRCFRNRCSWSLRNHGGQDCPWGSVCCKP	+31.16	+3	4188
hBD02/ Human	AF071216	DPVTCLKSGAICHPVFCPRRYKQIGTCGLPGTKCCKKP	+31.07	+6	4104

Brazilian pitvipers *Crotalus* (C.), *Bothrops* (B.), *Lachesis* (L.) and the colubrides *Phalotris* (P.), and *Thamnophis* (T.). The codifying sequences of genes were used to deduce the amino acid sequence and the Signal P software [74] used to determine the mature β-defensins. The hidrofobicity, net charge, and Molecular Weight were theoretical calculated using PepDraw software [http://pepdraw.com/ by Thomas C. Freeman, Jr. Accessed on 27 September 2021].

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
