# Peer review of "Antimicrobial Activity of Snake β-Defensins and Derived Peptides"

_toxins, 2021, doi:10.3390/toxins14010001_

Round 1
Reviewer 1 Report
Authors describe the antimicrobial activity of beta-defensin derivatives. This manuscript is well-organizaed and represent a interesting contribution to understand the structure-activity of cationic peptides derived from snakes.
Introduction is very extensive, containing aspects that does not contribute to support the hypothesis of the work. A reduction of some sections of introduction may be required.
Moreover, in Line 131, correct “highly” to “detailly”.
After these changes, this manuscript may be candidate to article for Toxins Journal.
Author Response
Dear Reviewer,
We appreciated the comments and suggestions.
The Introduction was improved, and the aspects that did not support the work were removed.
The Methods, Results, and Discussion were rewritten and improved.
The adjective at line 131 was changing accordingly.
Sincerely,
Reviewer 2 Report
I think this is a preliminary study. Moreover, the AMPs should be evaluated on resistant nosocomial bacteria.
Author Response
Dear Reviewer,
We appreciated the comments and suggestions.
The Introduction, Methods, Results, and Discussion were rewritten and improved.
Due to the weak antibacterial activity, we understood that it was not necessary to assay the defensins and peptides on nosocomial bacteria, so this assay was not performed.
Sincerely,
Reviewer 3 Report
In this manuscript, the authors selected 14 snake β-defensins, designed 6 derived peptides, and determined the antimicrobial activities of the peptides. Although the research subject is novel and interesting, the study is not appropriate designed and the results are too preliminary. Besides, the manuscript is not carefully prepared. There are too many errors in the manuscript and the results. Therefore, I think the manuscript in the present version is not suitable for publication in Toxins.
Specific comments:
- Why did the authors select the 14 β-defensins? They should give an explanation. Besides, why did they synthesize the β-defensins in linear structure? In fact, disulfide bond is important for the stable structure formation and activity exertion of β-defensins. Moreover, what’s the purpose of alkylation?
- The authors should describe the design strategy of the 6 derived peptides in more detail.
- The written of introduction and discussion sections is really in a too disordered manner. They should be carefully rewritten.
- The authors used 6 Gram-negative bacteria in the antibacterial activity assay. But in table 1 why did they show only two of them? Similarly, they also performed antifungal activity assay, where is the results?
- The authors discussed a lot about the cytotoxicity, salt sensitivity, and biocompatibility, why didn’t they perform these assays?
- Most of the content of 4.1. Molecular modeling and molecular properties calculation section should be moved to Results section. In fact, these calculations are meaningless.
- The tables were not well prepared. For example, what’s the exact unit of MICs in table 1? µM or µg/mL? they should be in consistent.
- There are too many grammatical errors in the manuscript. Some of them are really unacceptable.
Author Response
Dear Reviewer,
We appreciated the comments and suggestions.
- We choose crotamine, native and linear forms, and HBD-2 because they are classical defensins from snake and human origin. The others with gene described (Côrrea & Oguiura 2013 – pitviper sequences, Oliveira et al. 2018 – colubrid sequences), ranging different net charge, to test. The linear defensins were synthesized and tested in this study because previous studies showed that the folding structure is not crucial to antimicrobial activity. The alkylation avoids the SS bridge formation.
- The design strategy is described in more detail in Methods.
- The Introduction and the Discussion were rewritten and improved.
- On table 1, the four sensible bacteria are shown. Table 1 was divided into two, separating the antibacterial activity of defensins from the peptides. The result for antifungal activity was described only on the text in line 101, and since the samples were not effective on the assayed concentrations, we did not show it on a table.
- To avoid misunderstanding, we deleted the discussion about cytotoxicity, salt sensitivity, and biocompatibility.
- The figures of molecular modeling and properties were moved from Methods to Results.
- We improved the tables as requested.
The manuscript and the English grammar were reviewed.
Sincerely.
Round 2
Reviewer 2 Report
Although the authors try to improve the manuscript, I think this work is preliminary. So, more experiments are needed.
Author Response
Dear reviewer 2,
We appreciate the effort and time spent on the review.
This work is preliminary because the tested peptides and B-defensins did not show ideal activities; although the results were interesting, we thought it would be interesting to disclose. A newly revised manuscript, including suggestions made by the Academic Editor, was submitted.
Sincerely,
Reviewer 3 Report
None
Author Response
Dear Reviewer 3,
We appreciate the effort and time spent on the review. Your effort improved the manuscript a lot.
Sincerely,